# Genetic diversity affects ecosystem functions across trophic levels as much as species diversity, but in an opposite direction

Laura Fargeot[1,2]*, Camille Poesy[1], Maxim Lefort[1], Jerome G Prunier[1], Madoka Krick[1], Rik Verdonck[3], Charlotte Veyssiere[2], Murielle Richard[1], Delphine Legrand[1], Geraldine Loot[2], Blanchet Simon[1]*

[1]Centre National de la Recherche Scientifique (CNRS), Station d'Ecologie Théorique et Expérimentale, Moulis, France; [2]Université Paul Sabatier (UPS) Toulouse III, Toulouse, France; [3]Center for Environmental Sciences, Environmental Biology, Hasselt University, Diepenbeek, Belgium

**\*For correspondence:**
laura.fargeot@live.fr (LF);
simon.blanchet@sete.cnrs.fr (BS)

## eLife Assessment

This **important** study uses a comprehensive observational dataset to provide **solid** evidence on how genetic diversity and species diversity differentially affect multiple ecosystem functions within and across multi-trophic levels in an aquatic ecosystem. The work will be of interest to ecologists working on multi-trophic relationships and biodiversity.

**Abstract** Understanding the relationships between biodiversity and ecosystem functioning stands as a cornerstone in ecological research. Extensive evidence now underscores the profound impact of species loss on the stability and dynamics of ecosystem functions. However, it remains unclear whether the loss of genetic diversity within key species yields similar consequences. Here, we delve into the intricate relationship between species diversity, genetic diversity, and ecosystem functions across three trophic levels – primary producers, primary consumers, and secondary consumers – in natural aquatic ecosystems. Our investigation involves estimating species diversity and genome-wide diversity – gauged within three pivotal species – within each trophic level, evaluating seven key ecosystem functions, and analyzing the magnitude of the relationships between biodiversity and ecosystem functions (BEFs). We found that, overall, the absolute effect size of genetic diversity on ecosystem functions mirrors that of species diversity in natural ecosystems. We nonetheless unveil a striking dichotomy: while genetic diversity was positively correlated with various ecosystem functions, species diversity displays a negative correlation with these functions. These intriguing antagonist effects of species and genetic diversity persist across the three trophic levels (underscoring its systemic nature), but were apparent only when BEFs were assessed within trophic levels rather than across them. This study reveals the complexity of predicting the consequences of genetic and species diversity loss under natural conditions, and emphasizes the need for further mechanistic models integrating these two facets of biodiversity.

## Introduction

Diversity *within* and *among* species are both important to ensure and stabilize ecosystem functions (*Cardinale et al., 2012*; *Raffard et al., 2019*). Studies on the links between biodiversity and ecosystem

**eLife digest** When we speak about the loss of biodiversity, we often think of the loss of different species from an ecosystem. However, when ecosystems start to lose biodiversity, often, the first thing lost is diversity within species. This is, individuals of the same species become more like one another, leading to a loss of variety within a species.

This can cause issues at the species level as a lack of variation means that the species as a whole is less able to adapt to new environmental challenges, which can potentially lead to extinction. Humans are driving a loss of biodiversity worldwide, but it is unclear how the loss of diversity within a species affects ecosystems.

To answer this question, Fargeot et al. analyzed a complete food chain in an aquatic ecosystem in the wild, quantifying species diversity and using genetic tools to quantify within-species diversity. The researchers also quantified seven ecosystem functions associated with the ecosystem's productivity (how much biomass the ecosystem produces) and its ability to degrade dead organic matter.

Fargeot et al. found that the effects of losing within-species diversity in the ecosystem were as impactful as losing species diversity. The scientists also discovered that the relative impact of within- and between-species diversity on ecosystems were opposite. Losing species surprisingly increased the rate of ecosystem function, which also increased the amount of biomass produced and the amount of degraded organic matter. Conversely, losing diversity within species slowed down these ecosystem functions and thus decreased the services they can provide to humans.

These findings imply that measuring the loss of both within-species and between-species diversity is necessary to fully understand the effects of biodiversity loss. This will inform both conservation and agricultural efforts, where within-species diversity is often ignored.

functioning (BEFs) have primarily focused on the interspecific (species) facet of biodiversity (*Balvanera et al., 2006*; *Hooper et al., 2005*). However, the intraspecific (genetic) facet of biodiversity has also recently been shown to have substantial effects on ecosystem functions (*Crutsinger et al., 2006*; *Hughes and Stachowicz, 2004*; *Reusch et al., 2005*). Recent meta-analyses have shown that genetic diversity of plant and animal populations affect ecosystem functions, and that the magnitude (and shape) of intraspecific BEFs is similar to that of species diversity (*Raffard et al., 2019*; *Wan et al., 2022*).

Although natural assemblages encompass both intra- and interspecific diversity, most studies investigating BEFs are considering each biodiversity facet separately (but see, *Fridley and Grime, 2010*; *Prieto et al., 2015*; *Grele et al., 2024*). This makes it difficult to differentiate the relative role of genetic and species diversity in ecosystem functions, impeding general predictions regarding the consequences of biodiversity loss as a whole on ecosystem functions (*Blanchet et al., 2023*). For instance, we are currently unaware whether the loss of genetic diversity within a few species in an assemblage is as detrimental for ecosystem functions as a species loss, or whether the combined loss of genetic and species diversity may have non-additive consequences for ecosystem dynamics. Although these biodiversity loss scenarios are realistic, our knowledge on the relative role of genetic vs. species diversity in ecosystem functions are still too scarce to provide reliable predictions.

The few studies investigating the combined effects of genetic and species diversity on ecosystem functions were all conducted experimentally by manipulating the genetic and species diversity of assemblages under controlled conditions (*Fridley and Grime, 2010*; *Hargrave et al., 2011*; *Prieto et al., 2015*; but see *Grele et al., 2024*). Our understanding of genetic (intraspecific) and species (interspecific) BEFs therefore relies on simplified ecosystems that often lack variation in other factors (including spatial scales, abiotic factors, etc.), and in which feedbacks between ecosystem functions and biodiversity are limited (*Duffy et al., 2017*; *Prunier et al., 2023*). However, knowledge acquired from BEFs at the interspecific level reveals that environmental variation can either reduce or enhance the effects of biodiversity on ecosystem functions, hence generating large variance in the magnitude and direction of BEFs measured in the wild (*Hagan et al., 2021*; *van der Plas, 2019*). One can

therefore predict that, under natural conditions, the relative influence of genetic and species diversity on ecosystem functions may deviate from what has been quantified under controlled conditions, although it is difficult to predict the direction of this deviation as field studies (in particular for genetic BEFs) are too scarce to generate clear predictions. Therefore, we need further realistic field studies of BEFs, embracing the whole diversity of life forms (from genes to species) and across realistic environmental gradients to test whether – under natural conditions – species and genetic BEFs are of similar magnitude.

BEF studies often consider a single trophic level, despite accumulating evidence that biodiversity at a given trophic level can propagate across trophic levels, generating 'multi-trophic BEFs' (*Lefcheck et al., 2015*; *Soliveres et al., 2016*; *Seibold et al., 2018*). In particular, studies testing the joint effects of genetic and species diversity on ecosystem functions have mostly considered the effect of primary producer diversity on their own productivity ('*within-trophic level BEFs*', e.g. *Hargrave et al., 2011*; *Prieto et al., 2015*). However, genetic and species diversity within a given trophic level may have propagating effects on the ecosystem at other trophic levels (hereafter, '*between-trophic level BEFs*'). Indeed, it is predicted that a genetically diverse predator population shares their resources more efficiently than a genetically poor predator population, which might permit a higher prey species coexistence and hence a larger prey biomass (*between-trophic level* BEFs due to genetic diversity, e.g. *Raffard et al., 2021*). Alternatively, a species-rich community of primary producers likely exhibits higher primary production, as organisms in species-rich communities share basal resources more efficiently than in species-poor communities (*within-trophic level BEFs due to species diversity*, *Balvanera et al., 2006*; *Hooper et al., 2005*). Similarly, the relative impact of genetic and species diversity should be inconsistent across trophic. At higher trophic levels (e.g. predators), species richness is generally lower, which should increase the likelihood for genetic diversity (of a few species) to have strong effects on functions. A simple prediction might therefore be that the relative impact of genetic diversity on ecosystem functions should increase with increasing trophic levels (*Blanchet et al., 2020*). Studies considering genetic and species BEFs under a realistic multi-trophic scenario may thus help understanding the trophic contexts under which either genetic or species diversity is more impactful on ecosystem functions than the other, and to test whether genetic and species effects can propagate across trophic levels or not (*Seibold et al., 2018*; *Li et al., 2020*; *Moi et al., 2021*).

Here, we conducted a field study to test the relative importance of genetic and species diversity for ecosystem functions across multiple trophic levels in a natural landscape. We focused on three trophic levels from river ecosystems; riparian trees (primary producers), macroinvertebrate shredders (primary consumers), and fish (secondary consumers). For each trophic level, we quantified the species diversity of each community, as well as the genetic diversity of a single target and dominant species (*Alnus glutinosa*, *Gammarus* sp., and *Phoxinus dragarum* respectively). We further estimated several ecosystem functions, including leave decomposition of riparian trees, biomass (as productivity estimates) of each target species, and total biomass of each community within each trophic level. We relied on causal analyses, taking into account the direct and indirect effects of the environment (through biodiversity) on ecosystem functions (*Duffy et al., 2016*) to test (i) whether BEFs measured at the genetic level (*genetic* BEFs) are similar in magnitude and direction to BEFs measured at the species level (*species* BEFs); and (ii) whether *within-trophic level* BEFs are similar in magnitude than *between-trophic level* BEFs. We also tested whether the relative effects of species and genetic diversity on ecosystem functions (within or between trophic levels) are consistent across the three trophic levels (primary producers, primary consumers, and secondary consumers), in order to generalize findings along the trophic chain. We predicted that – contrary to what has been observed under controlled conditions – *genetic* BEFs and *species* BEFs will not be similar in magnitude, especially because environmental variation may modulate each of them differentially. We further expected that significant genetic and species BEFs will be observed both *within-* and *between-*trophic levels, leading to *within-* and *between-*trophic levels of similar magnitude. Finally, we predicted that the magnitude of *genetic* BEFs will be higher (than that of *species* BEFs) at the highest trophic level (secondary consumers) than at the lowest trophic level (primary producers), mainly because species richness at higher trophic levels presents a lower gradient than at the lowest trophic levels (*Figure 1* and *Figure 2*).

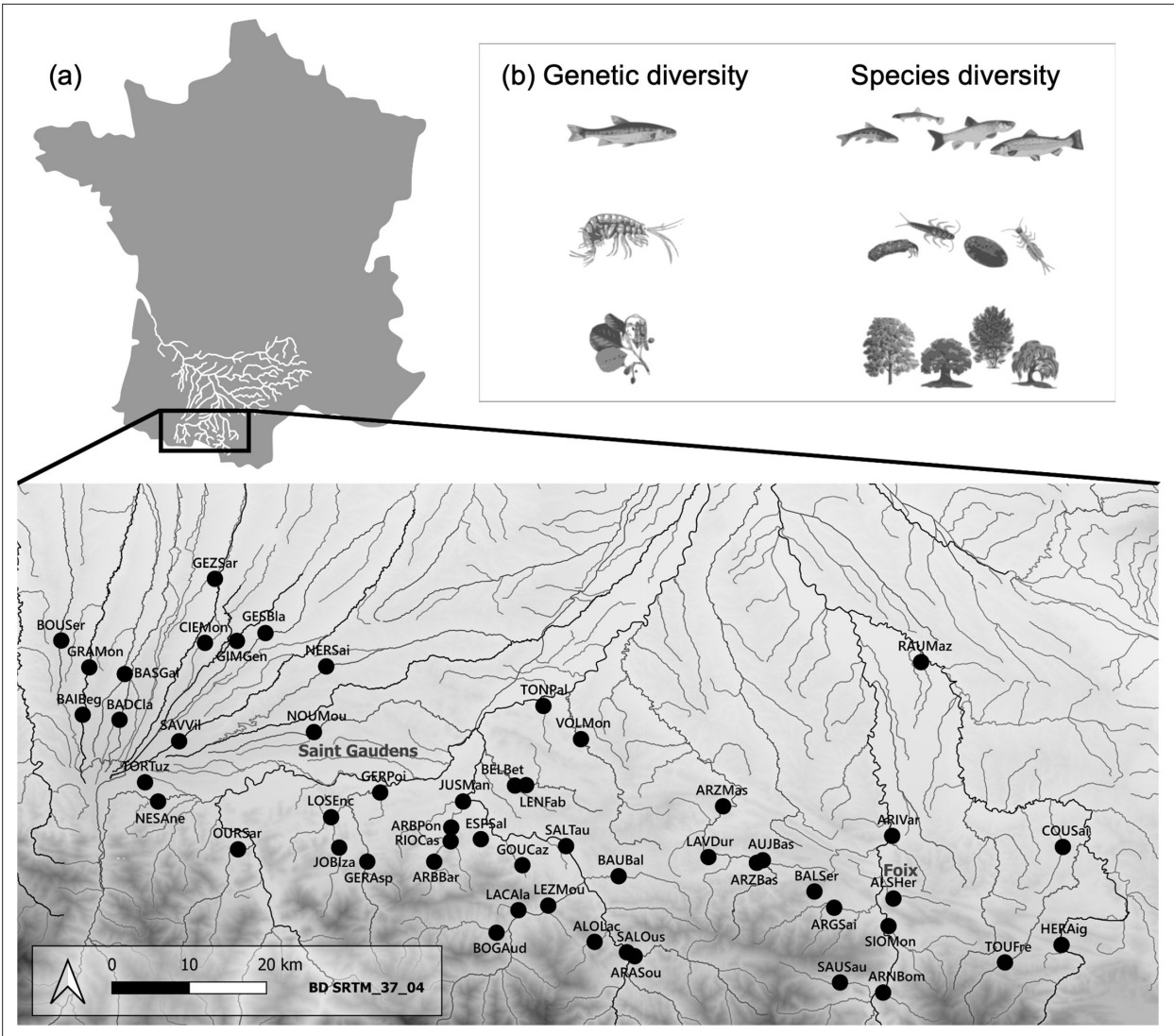

**Figure 1.** Location of sampling sites and illustration of the trophic chain. (**a**) Distribution of the 52 sampling sites (black dots) spanning an east-west gradient at the foothills of the Pyrenees Mountains (France). Each site is denoted by a six-letter code, with the first three letters indicating the river name and the last three letters indicating the closest city or village. (**b**) Our study focused on a tri-trophic food chain commonly found in mountain rivers, consisting of riparian trees, macroinvertebrate shredders, and fishes (from bottom to top). Within each trophic level, we measured two facets of biodiversity: genetic diversity in a single target species within each trophic level (specifically *P. dragarum*, *Gammarus* sp., and *A. glutinosa*), and species diversity from communities.

## Results

Details of causal models linking environmental parameters, species and genetic diversity, and ecosystem functions are graphically depicted in *Figure 3a–g*. Note that only relationships for which p-values were below 0.20 are shown on these graphs. This threshold was chosen arbitrarily to provide readable causal graphs and to highlight only on the most biologically relevant relationships.

The percentage of variance in ecosystem function explained by the environment and biodiversity varies from 10% (invertebrate biomass, *Figure 3e*) to 55% (*Phoxinus* biomass, *Figure 3f*) and was moderate overall. For all functions but the three biomass, part of the variance was (directly) explained by at least one out of the two environmental principal component analysis (PCA) axes. For some functions (e.g. *Phoxinus* biomass, *Figure 3f*), there was a combined effect of several biodiversity estimates, whereas for other functions (e.g. *Alnus* biomass, litter decomposition, *Figure 3a and c*) the effect of a single biodiversity estimate predominates. Overall, direct environmental effects on ecosystem functions did not predominate, and environmental effect sizes were similar (in strength) to

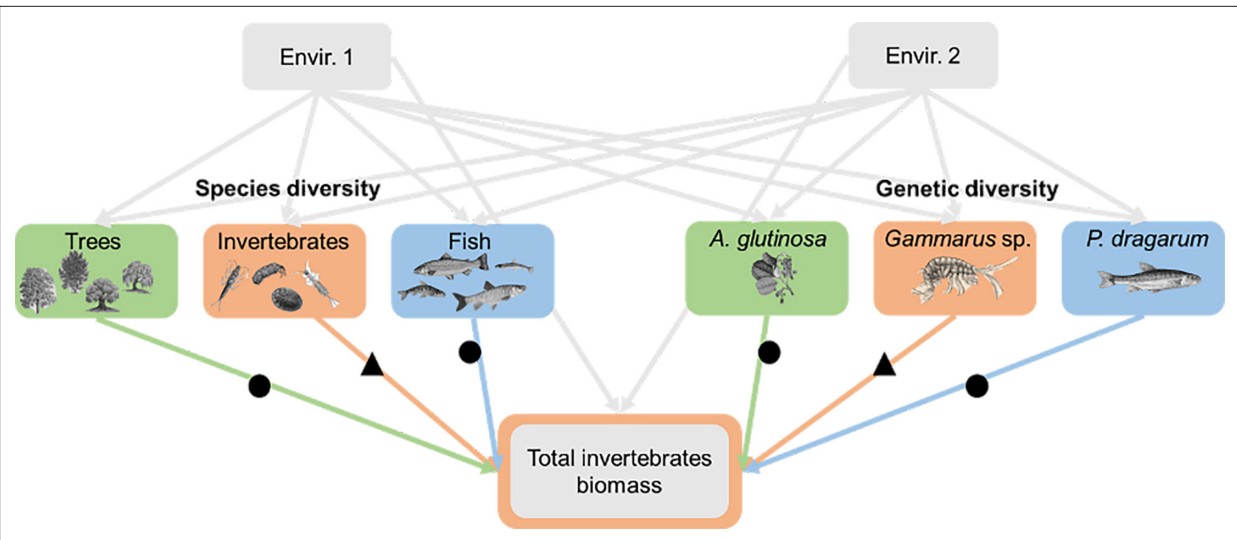

**Figure 2.** Example of one of the seven causal models used to quantify the relationships between (species and genetic) diversity and ecosystem functions. We focused on seven ecosystem functions associated with genetic and species diversity at three trophic levels (green for primary producer, orange for primary consumer, and blue for secondary consumer). Each relationship between biodiversity and ecosystem functions (n=6 values per function, but for some functions for which irrelevant links were not considered, see the text, n=32 values in total) was measured at the same trophic level (triangles) or at another trophic level (dots).

that of biodiversity effects, showing the non-negligible role of biodiversity for ecosystem functions in the wild.

Individual effect sizes (Zr) measured between BEFs (biodiversity and ecosystem functions relationships) were weak to moderate, irrespective of the considered ecosystem function and of the type of BEFs (*genetic/species BEFs*, *within-trophic level/between-trophic level* BEFs) (*Figure 4a and b*, *Supplementary file 2*). As expected under natural conditions (*Hagan et al., 2021*), BEFs ranged from negative to positive, and their distribution was centered around 0, although we observed a slight tendency for *genetic* BEFs toward positive values (*Figure 3b*). Only 4 out of the 34 BEFs were strong and significant; two significant BEFs concerned *species* BEFs (negative relationship between the biomass of *A. glutinosa* and the diversity of trees, Zr = –0.446, 95% CI [–0.695,–0.143]; negative relationship between the biomass of *P. dragarum* and the diversity of fish, Zr = –0.529, 95% CI [–0.802,–0.166]) and two concerned *genetic* BEFs (negative relationship between the biomass of *P. dragarum* and the diversity of *A. glutinosa*, Zr = –0.321, 95% CI [–0.602,–0.019]; positive relationship between the biomass of *Gammarus* sp. and the diversity of *P. dragarum*, Zr = 0.446, 95% CI [0.001, 0.829]) (*Figure 4a*). Noteworthily, for *within-trophic* BEFs, most case studies fall into the category whereby genetic BEFs tend to be positive and species BEFs tend to be negative (gray bottom-right square in *Figure 4a*).

We confirmed this visual tendency by summarizing all individual Zr through a meta-regression. Indeed, we found a significant interaction between the facet at which biodiversity is measured (genetic or species diversity), and the type of BEF that was measured (within- or between-trophic levels; *Table 1*; *Table 2*). This interaction indicates (i) that – overall – *within-trophic level* BEFs were significantly negative when considering species diversity (Zr$_{Within*Species}$ = –0.185, 95% CI [–0.343,–0.027]), whereas *within-trophic level* BEFs were significantly positive when considering genetic diversity (Zr$_{Within*Genetic}$ = 0.168, 95% CI [0.010, 0.326], see *Figure 5a*), and (ii) that this pattern was not observed for *between-trophic levels* BEFs, where no particular trend was observed (*Figure 5a*). Although most individual Zr were weak to moderate (and not significant), their consistency (in terms of magnitude and direction) resulted in a significant pattern whereby species and genetic diversity have opposite effects on ecosystem functions for *within-trophic level* BEFs; species diversity is negatively associated, whereas genetic diversity is positively associated with ecosystem functions, but only when the influence of biodiversity on ecosystem functions is measured within the same trophic level.

When including the trophic level at which biodiversity is measured, we found no significant interaction terms between trophic levels and other fixed effects nor any additive effect of trophic levels (see

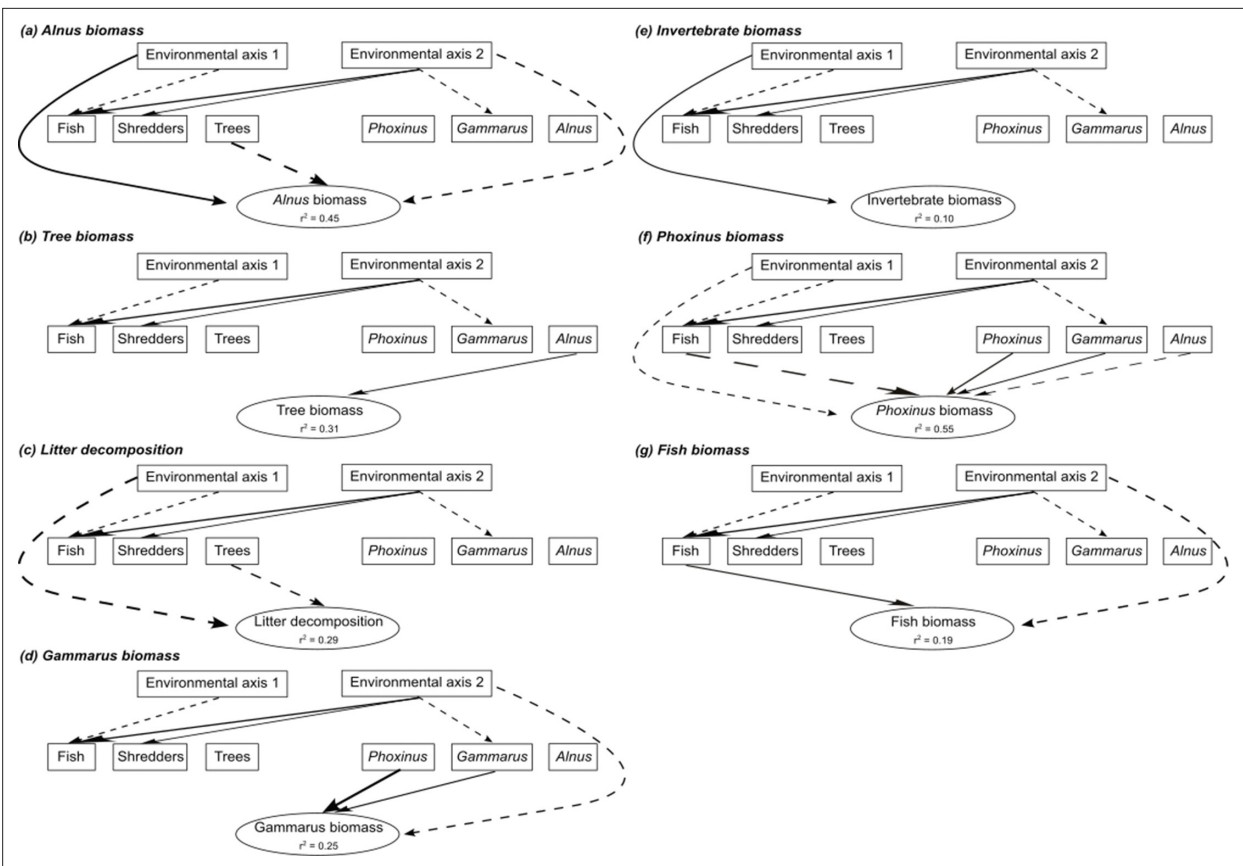

**Figure 3.** Details of the seven causal models linking abiotic parameters, species and genetic diversity, and ecosystem functions. Each causal graph (**a–g**) represents a simplified illustration of the relationships between the two principal component analysis (PCA) axes synthesizing the environmental parameters of each sampling site (Environmental axis 1 and 2), the species diversity estimated at each trophic level (boxes 'Fish', 'Shredders', and 'Trees'), the genomic diversity estimated from each focal species at each trophic level (boxes '*Phoxinus*', '*Gammarus*', and '*Alnus*'), and each ecosystem function (one model per function). Only the relationships for which the p-value was inferior to 0.20 are indicated for visual simplification. Full arrows indicated positive effects, whereas dotted arrows indicated negative effects. The width of the arrows is proportional to the size of their effects. The percentage of variance explained by environmental and biodiversity effects on ecosystem functions (r²) is indicated for each function.

*Supplementary file 1*). This indicates that our main findings were consistent across trophic levels, i.e., the respective negative and positive effects on ecosystem functions of species and genetic diversity hold statistically true across all trophic levels (*Figure 5b*).

## Discussion

We provide empirical evidence that, in natural ecosystems, the effect sizes of genetic and species diversity on multi-trophic ecosystem functions are of similar magnitude, but operate in opposite directions. Indeed, for BEFs measured within the same trophic level, the effects of species diversity across multiple ecosystem functions were moderately negative on average, whereas the effects of genetic diversity were moderately positive. This suggests an antagonistic effect between the genetic and the species components of biodiversity in the modulation of ecosystem functions within one trophic level. This antagonistic effect was not identified for BEFs measured across trophic levels, since in these cases the influence of both genetic diversity and species diversity across multiple ecosystem functions was generally not different from zero. These conclusions hold true across three trophic levels (plants, invertebrates, and fish), indicating that the relative effects of genetic and species diversity on ecosystem functions are not limited to a specific trophic level.

Our study is one of the few field-based study revealing BEFs across an entire (riverine) food chain spanning from primary producers to secondary consumers. Indeed, most previous BEF studies in the field focused on a single trophic level, and predominantly on terrestrial primary producers (*Duffy*

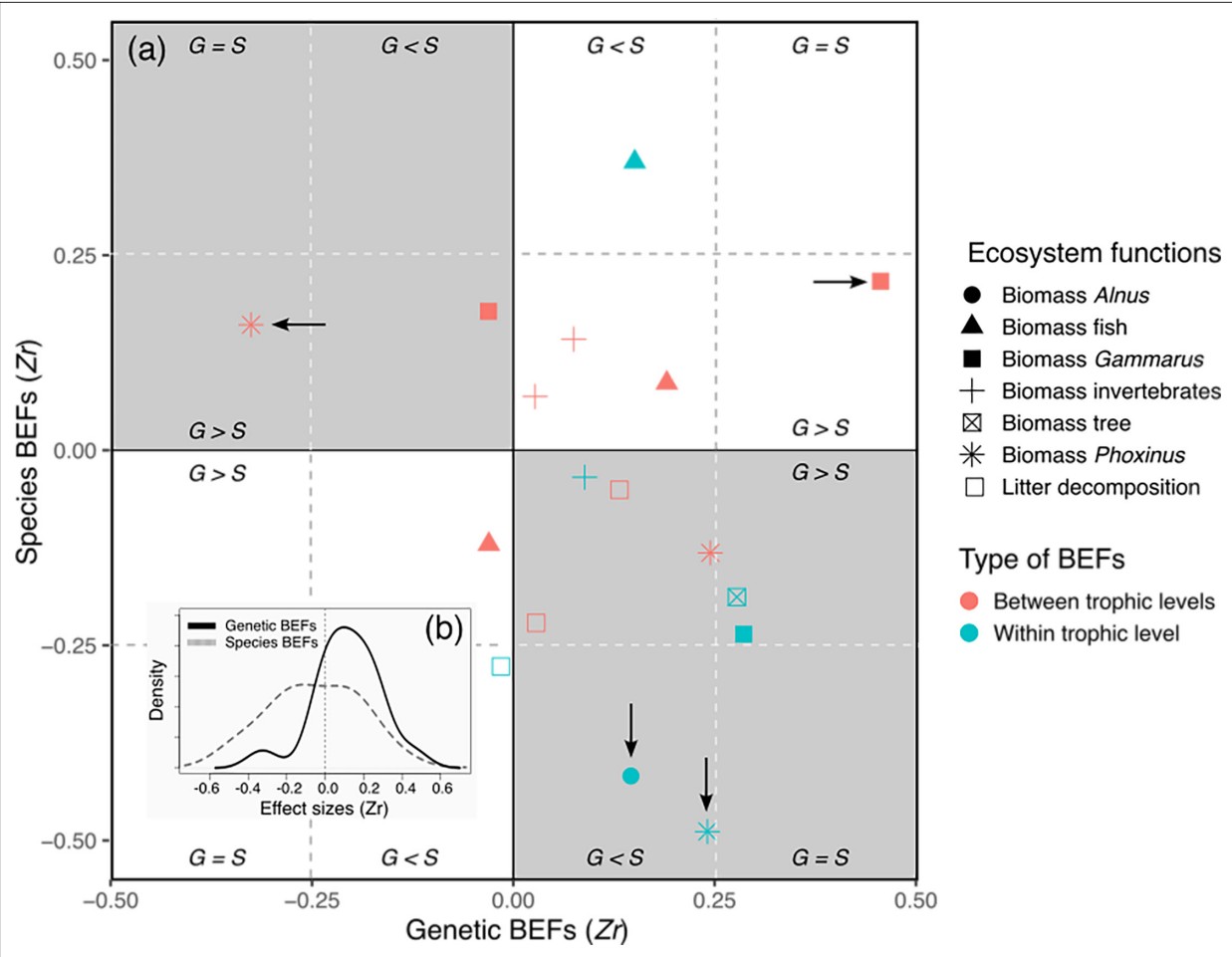

**Figure 4.** General description of individual effect sizes measured between biodiversity estimates and ecosystem functions (BEFs) in a riverine trophic chain. (**a**) The magnitude and direction of individual effect sizes (Zr) of biodiversity is shown for each ecosystem functions as a biplot between Zr associated with genetic diversity (y-axis, *genetic* BEFs) measured for one of three target species (*A. glutinosa*, *Gammarus* sp., and *P. dragarum*) and Zr associated with species diversity (x-axis, *species* BEFs) measured for one of three trophic levels (trees, invertebrates, and fish). For each ecosystem function (but the biomass of trees and of *A. glutinosa*), a total of six Zr are depicted in the biplot; four of them are associated with biodiversity measured at another trophic level than the one of the target functions (red symbols, e.g. effect of fish diversity on invertebrate biomass) and two of them are associated with biodiversity measured at the same trophic level than the one of the target functions (blue symbols, e.g. effect of fish diversity on fish biomass). The arrows indicate significant Zr (95% confidence intervals excluded 0, see ***Supplementary file 2***); vertical arrows are for significant *genetic* BEFs, horizontal arrows are for *species* BEFs. White quadrats stand for situation in which genetic and species BEFs are in the same direction, whereas gray quadrats indicate situation in which genetic and species BEFs are in the opposite direction. Within each quadrat, sub-quadrats indicate the relative magnitude of BEFs, i.e., whether *genetic* BEFs are stronger, weaker, or equal in magnitude than *species* BEFs. (**b**) Density plots displaying the distribution of individual Zr for *species* and *genetic* BEFs (dotted and full lines respectively).

*et al., 2017*; *van der Plas, 2019*, but see e.g. *Li et al., 2020*; *Moi et al., 2021*). This permitted encompassing a broad range of ecosystem functions that depict the overall functioning of a riverine ecosystem (rather than focusing on a single compartment). Moreover, we focused both on the effects of genetic and species diversity on these ecosystem functions, which has rarely (if not ever) been evaluated so far and which provides an exhaustive overview of BEFs in the wild. Our causal analyses also statistically took into account the direct (and indirect) effect of environmental factors on ecosystem functions, which is a prerequisite to isolate biodiversity effects. Nonetheless, causal relationships obtained from observational data (rather than from experimental data) are notoriously difficult to infer and must therefore be interpreted with care (*Duffy et al., 2017*). As a result, the BEFs we estimated display strong variability (ranging from negative to positive values) and a very few of them (4 out of 34) were statistically significant according to conventional thresholds. Although the statistical inferences made in this study are based on a large sample size, it is noteworthy that the general patterns we

**Table 1.** Characteristics of the first two principal components identified by the principal component analysis (PCA) ran on the 13 environmental variables.

The part of the total environmental variance (%) and the contribution of each variable on each component are shown. The variables that contributed significantly to the axis are highlighted in bold.

| | Component 1 | Component 2 |
|---|---|---|
| Part of total variance (%) | 21.66 | 16.37 |
| River width | 0.596 | 0.320 |
| Connectivity | –0.155 | **0.646** |
| Altitude | **0.648** | –0.385 |
| Distance from outlet | 0.528 | –0.331 |
| East-west gradient | 0.105 | **–0.795** |
| Oxygen concentration | **0.738** | 0.343 |
| Oxygen saturation | 0.266 | –0.509 |
| Water temperature | –0.594 | –0.012 |
| Specific conductivity | –0.463 | –0.045 |
| pH | 0.573 | 0.398 |
| Concentration in $NO_3^+NO_2$ | –0.369 | –0.286 |
| Concentration in $NH_4+$ | –0.019 | –0.207 |
| Concentration in $PO_4^{3-}$ | –0.279 | 0.236 |
| Global characteristic | Low altitude, poorly oxygenated site – high altitude, highly oxygenated sites | Poorly connected east site – highly connected west site |

will describe hereafter (and their interpretation) have to be considered with care, as we cannot rule out the possibility that some patterns might arise because of statistical biases rather than biological reality. Nonetheless, we – as ecologists – feel important to provide such a general picture from field data (even if partially distort by statistical limits), as this represents basic patterns that we have to understand.

We revealed that direct environmental effects on ecosystem functions were (in average) not stronger in intensity than biodiversity effects, which is coherent with previous syntheses on species BEFs in the wild (*Duffy et al., 2017*). Furthermore, environmental factors used to describe sampling sites in this study were not strong predictors of species and genetic biodiversity. Two non-exclusive hypotheses may explain this observation: (i) using PCA axes to resume environmental gradients may blur some specific environment-biodiversity links, and (ii) as shown and explained in a companion paper (*Fargeot et al., 2023*), the east-west gradient used in this study (rather than a classical upstream-downstream gradient) intrinsically limits the potential for strong environmental effects on biodiversity (which was the purpose of this sampling design). Nonetheless, after accounting for these environmental covariates, we found that most individual BEFs (either *genetic* or *species*, *within-trophic levels*

**Table 2.** ANOVA table for the linear mixed model testing whether the relationships between biodiversity and ecosystem functions (BEFs) measured in a riverine trophic chain differ between the biodiversity facets (species or genetic diversity) and the types of BEF (*within-* or *between-trophic levels*).

A Wald chi-square test is used to test the significance of each fixed effect.

| | Degree of freedom | Chisq-value | p-Value |
|---|---|---|---|
| (Intercept) | 1 | 0.287 | 0.595 |
| Biodiversity facet | 1 | 0.232 | 0.630 |
| Type of BEF | 1 | 5.393 | **0.020** |
| Biodiversity facet*Type of BEF | 1 | 5.567 | **0.018** |

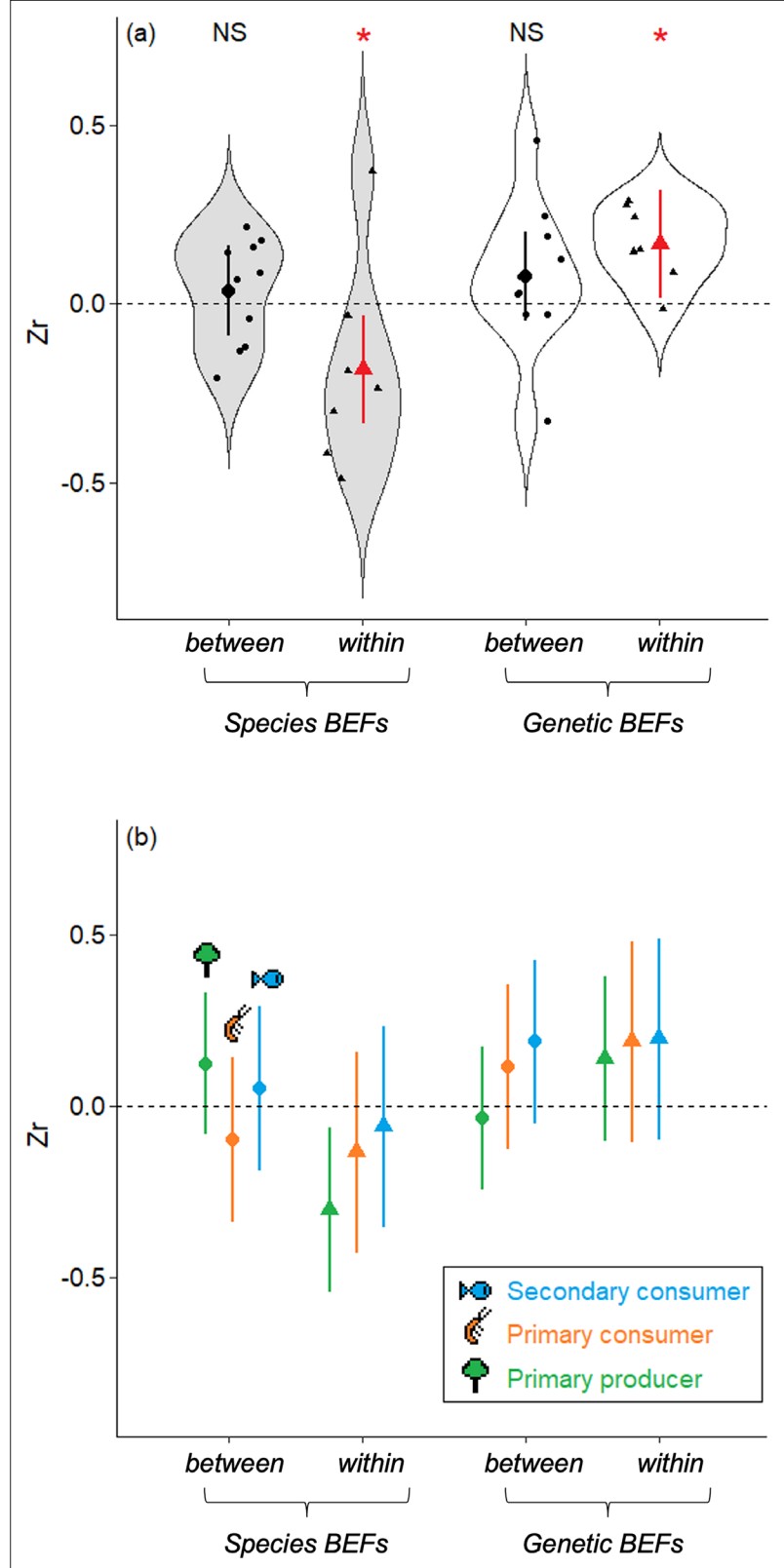

**Figure 5.** Magnitude and direction of the mean effects sizes estimated from the relationships between biodiversity and ecosystem functions (BEFs) measured in a riverine trophic chain. (**a**) The magnitude and direction of BEFs are expressed as effect sizes (Zr) and are displayed according to the facet used to measure biodiversity (genetic or species diversity, light gray and white boxplots respectively) and to the type of BEFs (*within-trophic level* BEFs or

*Figure 5 continued on next page*

*Figure 5 continued*

*between-trophic level* BEFs, triangles and dots respectively). Red color and stars indicate global effect sizes that are significantly different from zero (p-value<0.05). Large symbols are mean ± 95%IC estimated as marginal effects from the meta-regressions. Small symbols are raw estimates. (**b**) Same representation as (**a**) but with details at each trophic level (mean ± 95% IC estimated as marginal effects from the meta-regressions, green for primary producer, orange for primary consumer, and blue for secondary consumer). The trophic level at which BEFs are measured is coherent across all trophic levels.

---

or *between-trophic levels* BEFs) were weak to moderate in magnitude, and that they operated almost equally in both direction (i.e. positive and negative association between BEFs). As such, the distribution of individual effect sizes was centered around 0, for both *genetic* and *species* BEFs. Accordingly, there were only four individual BEFs that were significant, three out of them were negative and one was positive (*Supplementary file 2*). This general pattern (low to moderate BEFs with both positive and negative direction) is actually consistent with the most exhaustive meta-analysis having synthesized the magnitude and direction of *species* BEFs in the wild (*van der Plas, 2019*) and with recent conceptual works (*Hagan et al., 2021*) concluding that strong and positive BEFs should not be the norm in natural ecosystems, but rather that a mix of positive, neutral, and negative BEFs are expected. Our empirical findings are consistent with this conclusion.

We focused both on *within-trophic level* and *between-trophic level* BEFs, which likely encompasses a broad array of mechanisms sustaining potential associations between BEFs. For instance, two out of the four significant BEFs we reveal are negative association between species diversity (fish or tree species diversity respectively) and the biomass production of one of the target species (*Phoxinus* sp. and *Alnus* sp. respectively). These *within-trophic level* BEFs can – for instance – arise either because, if resources are limited, increased number of species within a patch limit the biomass production of each individual species, or because of a poorer competitive ability of the target species under some environmental conditions, which favors the settlement of additional species. Teasing apart these two hypotheses is difficult and further studies are needed to isolate underlying mechanisms. The other two significant BEFs concerned the association (either positive or negative) between the genetic diversity of a target species (*Alnus* sp. or *Phoxinus* sp.) with the biomass production of another target species (*Phoxinus* sp. or *Gammarus* sp., respectively). These *between-trophic level* BEFs likely arise through indirect effects implying the diversity and availability of (prey) resources. An obvious limit of this field-based study is the impossibility to tease out these mechanisms. Another limit is associated with the fact that, although environmental covariates were taken into account in causal models, they were synthetized by two PCA axes, and we cannot ensure that all potential environmental covariates have been taken into account (see above). This can influence the actual estimates of BEFs in the wild (*Duffy et al., 2017*). Nonetheless, it is noteworthy that we have previously shown that – in this dataset – species and genetic diversity were not correlated one to each other and that each biodiversity facet was sustained by different environmental predictors (*Fargeot et al., 2023*, see also *Figure 3*). This implies that environmental and biodiversity effects inferred in this study should not be strongly distorted by collinearities and can – in theory – be interpreted independently one from each other (e.g. the positive effect *Phoxinus* genetic diversity on *Phoxinus* biomass is independent from the negative of fish species diversity as the two estimates of biodiversity do not covary, see *Figure 3f*).

Keeping limitations associated with field-based studies into account (see above), revealing associations between ecosystem functions and species and genetic biodiversity (or the lack of) in natural ecosystems, is an important step forward to set theoretical and experimental approaches aiming at understanding this complex biological reality. Beyond individual BEF case studies (that were not the main aim of this study), their aggregation across trophic levels and biodiversity facets revealed a clear (and statistically supported) pattern whereby, within trophic levels, genetic and species diversity display antagonistic association with ecosystem functions; the global effect of species diversity across multiple ecosystem functions was negative, whereas the global effect of genetic diversity was positive. This pattern emerges from the 'cumulative' effects of weak to moderate associations between BEFs that consistently point toward the same direction (positive for genetic diversity, negative for species diversity), emphasizing the meaningfulness of meta-regressions (and more generally approaches based on effect sizes rather than on p-values) to reveal biological patterns. We hereafter discuss the ecological relevance of this general pattern.

Our results confirm a previous meta-analysis demonstrating that genetic and species diversity modulates ecosystem functions with a similar magnitude (*Raffard et al., 2019*), and results from few experimental studies that manipulated both the genetic and species components of biodiversity under controlled conditions (e.g. *Jiang et al., 2022*; *Prieto et al., 2015*). Indeed, within trophic levels, the absolute mean effect size of genetic and species diversity across ecosystem functions were of the same magnitude (|Zr|=0.168 and 0.185 for genetic and species diversity effects respectively), and slightly greater than the effect sizes reported under controlled conditions (|lnRR|=0.132 and 0.134 for genetic and species diversity effects respectively, *Raffard et al., 2019*) and those more generally reported for *species* BEFs (|Zr|=0.101, *Balvanera et al., 2006*). Although comparing effect sizes among studies that strikingly differ in their spatial coverage (small or large spatial scale), their taxonomic focus (e.g. primary producers vs. predators, species vs. genetic diversity, etc.) and/or their approaches (experimental vs. observational studies) is questionable (especially given the non-linear nature of BEFs), our findings suggest for the first time that under natural conditions, the effects of genetic and species components of biodiversity on ecosystem functions are comparable. However, our study goes two steps further as (i) it extends the conclusion made by *Raffard et al., 2019*, to multiple trophic levels and (ii) it suggests that the effects of genetic and species BEFs can actually operate in opposite directions.

As pointed out by *Raffard et al., 2019*, the vast majority (91% of 23 reviewed studies by 2019, see also *Wan et al., 2022*) of studies investigating the effects of genetic diversity on ecosystem functions have focused on primary producers, and all of them were based on experiments, which is also the case for most studies manipulating both genetic and species diversity. These trends strongly hamper any generalization. On the contrary, our findings provide a solid support for broadening the conclusion that both genetic and species diversity can influence ecosystem functions in the wild. More strikingly, our results suggest that, although the absolute effect sizes of genetic and species BEFs are of similar magnitude, for *within-trophic level* BEFs, the direction of their effects is opposite; species diversity (in general) reduces the rate of ecosystem functions, whereas genetic diversity enhances the same functions. For instance, all other things being equal, higher fish species diversity is associated with a lower productivity (biomass) in *P. dragarum* (see above for potential explanations), whereas its own genetic diversity tends to be associated with a higher productivity (*Supplementary file 2*). In this specific case, genetic and species diversity of the same trophic group (fish) tended to have opposite effects on the same function (productivity of *P. dragarum*). However, in most cases this was not the case as genetic diversity was positively associated with some functions, whereas species was negatively associated with other functions. The distinction between these two patterns is important as in the latter case (genetic and species diversity are associated with different functions) managing/conserving the intra- and interspecific diversity of a single trophic group (e.g. trees) can alter more than one ecosystem function, and sometimes functions that are even not directly associated to the managed trophic group. Moreover (and importantly), as genetic and species diversity have been found to be uncorrelated spatially in this landscape (*Fargeot et al., 2023*), covariation among diversity estimates cannot explain these patterns. These antagonistic effects of genetic and species diversity on ecosystem functions parallel previous experimental findings on plants (*Hazard et al., 2017*; *Tang et al., 2022*). It is now essential to understand the mechanisms sustaining these antagonistic effects as a step forward.

S*pecies* BEFs were on average negative (see *Supplementary file 2* for individual estimates), which contrasts with the general view that species biodiversity favors ecosystem functions, although it is not that surprising (*Dee et al., 2023*; *Hagan et al., 2021*). Indeed, the net effect of species biodiversity on ecosystem functions results from the combined effects of both negative factors, arising from antagonistic interactions such as negative complementarity or negative selection effect, and positive factors, arising from beneficial interactions such as niche complementarity or facilitation (*Loreau and Hector, 2001*). We can speculate that, in our case, the net effect of interspecific interactions mostly results from negative complementarity among species (or strong negative selection effect), whereas the net effect of intraspecific interactions may result from facilitative interactions and/or improved niche complementarity with increased genetic diversity. Intraspecific competition is generally stronger than interspecific competition (*Connell, 1983*), and intraspecific interactions could be expected to lead more frequently to negative complementarity (and hence negative *genetic* BEFs) than interspecific interactions. Since we observe the opposite, we can hypothesize that genetic diversity is essential

to increase niche complementarity within species (*Bolnick et al., 2003*) and hence to reduce the pervasive effects of intraspecific interactions (*Hughes et al., 2008*; *Prunier et al., 2023*). Given the empirical nature of our study and the fact that our meta-regressive approach includes several types of BEFs (e.g. species richness acting either on the biomass of a single focal species or on the biomass of an entire focal community), it is hard to tease apart specific and underlying mechanisms. Theoretical approaches, modeling simultaneously the genetic and species components of biodiversity, would be extremely useful to reveal the mechanisms sustaining opposite effects of intra- and interspecific diversity on ecosystem functions.

These antagonistic effects were observed only for BEFs measured *within* trophic levels, not for those measured *between* trophic levels. An overall *between-trophic level* BEF not different from zero suggests that biodiversity at a trophic level has only limited impact on ecosystem functions at another trophic level. For example, the biomass of *P. dragarum* was primarily influenced by genetic and species diversity in fishes, rather than the diversity of their preys (*Supplementary file 2*). However, for both genetic and species estimates of biodiversity, there was a substantial variation in effect sizes for *between-trophic level* BEFs that ranged from negative to positive BEFs (*Figure 5*). This suggests that biodiversity effects across trophic levels may be more variable in their direction than within-trophic level BEFs, which appear as more constrained. Variability in the magnitude and direction of effect sizes for *between-trophic level* BEFs likely blur a more general trend, but this variation is actually expected under natural conditions in which interactions involve multiple prey and predator species, fostering co-adaptation among communities from different trophic levels (*Aubree et al., 2020*; *Poisot et al., 2013*). In these cases, trophic complementarity between two trophic levels (i.e. the originality of a species based on the identity of the species it interacts with) might be a stronger determinant of ecosystem functions than complementarity measured at either one of the two trophic levels (*Poisot et al., 2013*). Quantifying trophic complementarity among our three target species (and communities) using stable isotope or gut content analyses for instance would be extremely valuable to assess whether this complexity can better explain BEFs between trophic levels than diversity measured at one of the trophic level (*Aubree et al., 2020*).

The empirical patterns we revealed here were all extremely consistent across the three trophic levels, hence allowing generalization. It is noteworthy that, although statistically strong and consistent, these patterns must be interpreted with care as field-based approaches are limited in properly taking into account the environmental heterogeneity of natural ecosystems (*Hagan et al., 2021*). BEFs were not particularly stronger at any specific trophic level and the relative effects of genetic and species diversity were not dependent on the trophic level at which the function was estimated. We may have expected a stronger top-down regulation (i.e. biodiversity of predators has more effects than biodiversity of preys) of ecosystem functions since previous studies showed that biodiversity loss should have greater consequences for multi-functionality when it occurs at higher trophic levels (*Lefcheck et al., 2015*; *Seibold et al., 2018*). For instance, increased genetic diversity within a predatory fish species has experimentally been shown to indirectly increase the rate of litter decomposition by increasing the diversity of shredders (*Raffard et al., 2021*). Similarly, the relative effects of genetic and species diversity on functions may have varied among trophic levels, and in particular the relative importance of genetic diversity may have been higher for species-poor trophic levels (i.e. fish community) because of a 'compensatory effect'. We found no evidence for these potential trophic-level dependencies, but instead found extremely consistent patterns, which, from a broader perspective, reveal the importance of integrating both multi-trophic and multi-faceted approaches in predicting the overall consequences of biodiversity loss on ecosystem functioning.

To conclude, we found that the genetic (intraspecific) and species (interspecific) facets of biodiversity are both important drivers of multiple ecosystem functions in a natural and multi-trophic context. In the wild, these two facets of biodiversity can, as expected, generate low to moderately high impacts on ecosystem functions measured across three trophic levels, and they can operate in opposite directions (but on different functions; genetic diversity is positively associated with some functions, species diversity is negatively associated with other functions). This shows the importance for managers to develop integrative conservation plans spanning the entire diversity of life (from genes to species). For instance, genetic diversity loss often precedes species loss, and our results suggest that – in mountain streams – losing genes may actually be particularly detrimental for the performance of ecosystem functions. As such, it appears essential to maintain populations with high levels of genetic

diversity in these ecosystems. Future studies should (i) extend these findings to other ecosystems and by quantifying natural genetic variation in more than a single species per trophic level, (ii) generate theoretical predictions regarding the mechanisms sustaining the antagonistic effects of genetic and species diversity on functions we revealed, and (iii) use a broader integrative approach for estimating biodiversity across facets (*inclusive* biodiversity) by using either a trait-based approach or a genetic-based approach as recently proposed by *Blanchet et al., 2023*, and *Loreau et al., 2023*.

## Materials and methods
### Sampling sites and trophic chain
We sampled 52 sites in Southern France from the Adour-Garonne watershed, and distributed along an east-west gradient in the Pyrenees Mountains (*Figure 1a*, *Blanchet, 2024*). We acquired data on species diversity, genetic diversity, and ecosystem functions at three trophic levels (primary producers, primary consumers, and secondary consumers) (*Figure 1b*). Riparian trees (57 species in the sampled area) provide organic matter in the form of fallen leaves as a food source for decomposers. We selected the common alder *A. glutinosa* for acquisition of genetic data due to its dominance at most sites and its functional relevance, as its roots serve as shelters for many aquatic species and are involved in nitrogen fixation. Macroinvertebrate shredders (101 genera in the sampled area) are primary consumers using leaves as resources, and converting them into accessible organic matter for other species. We focused on the most abundant Gammarid (Crustacean) species for genetic data acquisition, referred to as *Gammarus* sp. This species has not yet been formally named although it is phylogenetically distinct from its closest relative, *Gammarus fossarum* (*Carnevali, 2022*; Piscart, unpublished data). This species is particularly efficient at decomposing tree leaves, in particular those from *Alnus* (*Macneil et al., 1997*). Fish (20 species in the sampled area) are secondary consumers feeding on invertebrates (among others). We used the minnow *P. dragarum* as the fish target species as it is an abundant and important predator strongly impacting invertebrate communities (*Raffard et al., 2021*).

### Biodiversity estimates
#### Species datasets
At each site, we collected data on the abundance of all species within each trophic level, at one occasion for trees (July-August 2021) and two occasions for invertebrates (July and November 2020) and fishes (mid-July to mid-August 2020 and 2021), to obtain accurate biodiversity estimates. We identified tree species along a 200 m transect of each river bank, excluding trees with trunk smaller than 2 cm in diameter and more than 1 m away from the bank. The abundances of trees were estimated as the total number of individuals per species and per site. For invertebrates, we identified shredders to the genus level (or to the family level for some groups such as chironomids) sampled from two types of standardized traps installed in four micro-habitats distributed along the 200 m transect used to identify trees: natural coconut brushes (15*5.5 cm, bristles length 7.5 cm) recovered after 1.5 month of colonization, and litter bags (15*11 cm, 0.8 cm mesh size) filled with senescent *Alnus* leaves from each site and recovered after 9 days of colonization (see below). We calculated abundances of each genus by summing the number of individuals per genus found in the coco brushes and the litter bags, and we averaged the abundances over the two sampling occasions to get a single estimate per genus per site. For fish, we collected all specimens during single-pass electric fishing sessions over a mean area of ~469.9 m$^2$ (±174 m$^2$) distributed along the 200 m transect. We anesthetized, identified, and counted individuals at the species level. We calculated fish abundances as the number of individuals per species and per m$^2$, and we averaged the abundances over the two sampling occasions as for invertebrates. Fish species number varies from 1 to 11, invertebrate genus number varies from 15 to 42, and the tree species number varies from 7 to 20 (see *Fargeot et al., 2023*, for details).

#### Genetic datasets
At each site, we collected tissue from up to 32 individuals of each of the three target species, a sample size having found sufficient for estimating the genomic diversity of populations (*Hale et al., 2012*). We collected fresh leaves of *A. glutinosa* in May 2020, specimens of *Gammarus* sp. in February 2020, and a piece of pelvic fin from *P. dragarum* individuals in summer 2020. The DNA of these samples

was extracted using commercial kits for *Alnus* and *Gammarus* sp. and a salt-extraction protocol for *P. dragarum* (see *Fargeot et al., 2023*, for details). For each specimen, DNA concentrations were measured using Qubit 3.0 fluorometer (Life Technologies, USA). Sequencing was performed based on equimolar pools of DNA ('pool-seq' approach, *Schlötterer et al., 2014*) from each population and each species. For *Gammarus* sp., we also obtained an ~600 bp mitochondrial sequence from the COI mitochondrial gene from each individual to ensure identification and avoid mixing individuals from different species. *Gammarus* sp. was found allopatric in most sites, but for a few sites from the eastern part of the area in which two species were identified (*Carnevali, 2022*). In this latter case, we conserve only the target species for creating the DNA pools. We created one DNA pool per site per species (52 pools for *A. glutinosa*, 47 pools for *Gammarus* sp., and 44 pools for *P. dragarum*) and performed double-digest restriction site-associated DNA sequencing for *A. Glutinosa* and *Gammarus* sp. (respectively, PstI/MseI and Pst/HindIII enzymes) and normalized genotyping-by-sequencing for *P. dragarum* (MsII enzyme). Library preparation and pool-sequencing were executed by LGC Genomics (Biosearch Technologies, Germany) on an Illumina NovaSeq (2×150 pb). Data processing was performed following *De Kort et al., 2018*, except that read mapping was performed on reference genomes. The genome of *A. glutinosa* was already available (*Griesmann et al., 2018*), and we assembled reference genomes from Illumina short-read sequencing and PacBio long-read sequencing for *Gammarus* sp. (available upon request) and *P. dragarum* (accession number on DDBJ/ENA/GenBank: JARPMJ000000000), respectively. SNP calling was performed with (i) filtering of raw sequencing files; (ii) indexing of reference genomes; (iii) mapping reads to the reference; (iv) filtering for unpaired and badly/non-mapped reads; (v) assembling all read information in a single file per population and per species; and (vi) calculating SNP allelic frequencies (*De Kort et al., 2018*). The total numbers of SNPs retrieved were 583,862 for *A. glutinosa*, 331,728 for *Gammarus* sp., and 414,213 for *P. dragarum* (see *Fargeot et al., 2023*, for details).

## Species and genetic diversity estimates

We calculated α-diversity per site using the Shannon entropy from the 'hillR' R package for both species and genetic diversity. The Shannon entropy is a metric of evenness that takes into account the distribution of allele or species abundances within each site (*Chao et al., 2014*) by weighting each species/allele by its proportional abundance (q=1). Results were similar when using the Simpson's diversity index (q=2, results not shown). It is noteworthy that – given the spatial extent of the sampling area and the number of sampling sites – genetic and species diversity estimated in this study constitutes a fair representation of the biodiversity found in the rivers from the Pyrenean Piedmont, covering a wide range of biological complexity.

## Ecosystem function measurements

At each site, we measured seven ecosystem functions. We collected biomass production data of all species at each trophic level (hereafter 'total biomass') and the biomass production of each target species as estimates of productivity, as well as the decomposition rate of *Alnus* leaves. Productivity – as we quantified it – is obviously affected by local environmental characteristics, and for this reason, we took into account these potential environmental effects (see hereafter). For riparian tree biomass, we used the trunk diameter of each single tree as a proxy of individual tree biomass, and we summed the trunk diameters of all trees found along the transect (divided by the length of the transect) to estimate the total tree biomass per site and per meter of bank. The same approach was used to estimate *A. glutinosa* biomass. For macroinvertebrate shredders, we estimated the total invertebrate biomass by drying all individuals for 24 hr at 60°C before weighing them ($10^{-4}$ g precision). The same procedure was used to estimate the biomass of *Gammarus* sp. For both estimates, we averaged biomasses over the two sampling sessions. For fish, (fresh) total fish biomass was estimated as the total weight of all individuals (0.01 g precision) per site, whereas *P. dragarum* biomass was the mass of all *P. dragarum* specimens per site. Fish biomasses were averaged over the two sampling sessions.

For the decomposition rate, we quantified leaf mass loss in litter bags placed in four micro-habitats per site twice (July and November 2020). We gathered and dried senescent leaves during fall 2019 from five *Alnus* trees per site to limit individual-specific effects on decomposition. Litter bags were 15 cm × 11 cm pockets of plastic-wire mesh (mesh size; 8 mm to allow invertebrates colonization) in which we introduced 4 g of dried leaves before closing the bags with staples. We installed three bags

per micro-habitat (12 per site) that we removed sequentially after ~9 days, ~18 days, and ~27 days respectively to estimate decomposition rates. Bags were brought back to the laboratory, the remaining leaves were cleaned, dried, and weighed. Decomposition rate was estimated as the slope of leaf mass loss over time (obtained from a linear model) that we averaged across replicates and temporal sessions (*Raffard et al., 2021*).

## Environmental data

A major challenge for inferring BEFs from empirical data is to take into account the direct and indirect (through biodiversity) effects of environmental factors on ecosystem functions (*Duffy et al., 2016*; *Duffy et al., 2017*). Failing to do this may result in overestimated and/or artifactual BEFs, especially if the same environmental factor simultaneously affects biodiversity and ecosystem processes (*Grace et al., 2016*). For each site, we measured 13 variables related to river topography and physico-chemical characteristics that likely influence biodiversity and ecosystem processes (*Altermatt, 2013*). *River bed width* (m) was averaged from five measurements per site. *Connectivity* was calculated as the 'closeness centrality', i.e., the inverse of the sum of the distances of a node to all other nodes along the shortest paths possible (*Altermatt, 2013*), using QGIS and the 'RiverDist' R package. *Altitude* (m), *distance from the outlet* (m), and *east-west gradient* (longitudinal position along the Pyrenees chain) were measured using QGIS; *oxygen concentration* (mg/L), *oxygen saturation* (%), *water temperature* (°C), *specific conductivity* (μS/cm), and *pH* were measured (and averaged) in summers 2020 and 2021 using a multi-parameter probe (Aqua TROLL 500, In-Situ Inc). Concentration of $NO_3$-, $NO_2$-, $NH_4^+$, and $PO_4^{3-}$ were estimated (and averaged) during summers 2020 and 2021 from a filtered water volume (100 mL) using the Alpkem Flow Solution Iv Autoanalyzer (OI Analytical).

A PCA combining all 13 variables was performed using the R package 'ade4' (*Dray and Dufour, 2007*), and coordinates of each site on the two first axes (38.03% of the total variance, see *Table 1*) were used as two synthetic environmental variables for further analyses. We kept only these two first axes to avoid collinearity and over-parameterization of subsequent models. The first axis is defined by a strong contribution of (in decreasing order) oxygen concentration and altitude (*Table 1*). The second axis is defined by a strong contribution of east-west gradient and connectivity (*Table 1*).

## Statistical analyses

### BEF relationships

To quantify the magnitude of association between BEFs, we performed piecewise structural path models (pSEM, 'piecewiseSEM' package, *Lefcheck, 2016*). pSEM allows modeling direct and indirect causal relationships among a set of response variables and predictors (*Shipley, 2009*). Further, pSEM uses local estimates of each linear structural equation separately (i.e. parameters are estimated from a series of independent models forming a general causal graph), which allows the inclusion of a large number of parameters despite modest sample sizes (*Shipley, 2009*). We ran a pSEM for each ecosystem function separately (i.e. seven pSEMs, see an example in *Figure 2*). In each pSEM, the ecosystem function was the dependent variable whereas the six biodiversity estimates (species and genetic diversity estimated for each trophic level) and the two synthetic environmental variables were the predictors. In each model, environmental predictors were allowed to explain each biodiversity estimate (indirect effects of environmental variables through their influence on biodiversity, see *Figure 2*). For some functions (in particular those associated with plant biomass), irrelevant biodiversity-functions links were not included (e.g. the impact of fish or invertebrate diversity on tree biomasses), which results in 34 BEFs (out of the 42 possible links) having been included in the meta-regression (see hereafter).

From each pSEM, we retrieved the local parameter (standardized estimate, an equivalent to a coefficient of correlation) associated with the direct effect of each biodiversity estimate (six per function, but for some functions for which ecology-irrelevant BEFs were excluded) on the function (colored arrows in *Figure 2*), which provides both the magnitude and the direction of each BEF. To smoothen comparison, we calculated a standardized effect size for each BEF by applying the Fisher's Z transformation (Zr) to the standardized estimates. Positive Zr indicate positive associations between BEFs, whereas negative Zr indicate negative relationships. The higher the absolute value of Zr, the higher the strength of the association. Zr therefore indicate both the direction (positive or negative) and the

magnitude of the associations. Our seven measures of ecosystem functions were not correlated one to each other (all $r_{Pearson}<|0.39|$).

## Direction and magnitude of all types of BEFs

We used a linear mixed model to test (i) whether the magnitude and direction of *genetic* BEFs are similar to those of *species* BEFs, and (ii) whether *within-trophic level* BEFs are similar in effect size to *between-trophic level* BEFs. In this model, Zr (providing the direction and magnitude of each BEF, n=34) was the dependent variable, and the predictors were the diversity facets used to measure biodiversity (genetic or species diversity) and the type of BEF (*within-trophic* or *between-trophic* levels, triangles vs. dots in *Figure 2*). We included the two-term interaction between diversity facet and type of BEF to test whether the magnitude and direction of *genetic* and *species* BEFs are consistent across *within-trophic level* and *between-trophic level* BEFs. We further included in this model the type of ecosystem function as a random term (to take into account that each ecosystem function was associated with several biodiversity estimates) as well as the inverse of the asymptotic variance ($v_z$ = n–3) associated with each effect size as a weighting parameter for each case study (*Balvanera et al., 2006*; *Raffard et al., 2019*).

We ran an additional linear mixed model similar to the previous one, except that we added as a fixed effect the trophic level at which biodiversity was measured to estimate BEFs (primary producers, primary consumers, or secondary consumers) as well as all interaction terms. Interaction terms allow testing the consistency of major conclusions across trophic levels, thereby determining the extent to which our findings can be generalized along the trophic chain. Models were run using the lmer function ('lme4' package) and significance of fixed effects was determined using type III ANOVA with Wald chi-square tests (function ANOVA from the 'car' R package, α=0.05).

# Additional information

### Funding

| Funder | Grant reference number | Author |
|---|---|---|
| Agence Nationale de la Recherche | ANR-18-CE02-0006 | Blanchet Simon |
| Agence Nationale de la Recherche | ANR-10-LABX-0041 | Blanchet Simon |

The funders had no role in study design, data collection and interpretation, or the decision to submit the work for publication.

### Author contributions

Laura Fargeot, Conceptualization, Resources, Data curation, Software, Formal analysis, Supervision, Validation, Investigation, Visualization, Methodology, Writing – original draft, Writing – review and editing; Camille Poesy, Maxim Lefort, Madoka Krick, Rik Verdonck, Charlotte Veyssiere, Murielle Richard, Investigation, Methodology, Writing – review and editing; Jerome G Prunier, Formal analysis, Methodology, Writing – review and editing; Delphine Legrand, Geraldine Loot, Conceptualization, Supervision, Writing – original draft, Writing – review and editing; Blanchet Simon, Conceptualization, Formal analysis, Supervision, Funding acquisition, Validation, Investigation, Methodology, Writing – original draft, Project administration, Writing – review and editing

### Author ORCIDs

Blanchet Simon  https://orcid.org/0000-0002-3843-589X

Joint Public Reviews: https://doi.org/10.7554/eLife.100041.4.sa1
Author response https://doi.org/10.7554/eLife.100041.4.sa2

# Additional files

## Supplementary files

Supplementary file 1. ANOVA table for the linear mixed model testing whether the relationships between biodiversity and ecosystem functions (BEFs) measured in a riverine trophic chain differ between the biodiversity facets (species or genetic diversity), the types of BEF (*within-* or *between-trophic levels*), and the trophic levels at which BEFs are estimated (primary producers, primary consumers, or secondary consumers). A Wald chi-square test is used to test the significance of each fixed effect.

Supplementary file 2. Estimates of individual effect sizes of biodiversity and ecosystem functions (BEFs) (Zr, n=34) for each ecosystem function, each biodiversity facet (genetic or species diversity), and each type of BEF (within- or between-trophic levels). 95% confidence intervals are provided together with the estimate of each BEF. BEFs are considered as significant when the 95% CI does not overlap 0. p-Values estimated from t-test are also provided.

MDAR checklist

## Data availability

All data generated or analysed during this study are available at https://doi.org/10.6084/m9.figshare.25392496.v1.

The following dataset was generated:

| Author(s) | Year | Dataset title | Dataset URL | Database and Identifier |
| --- | --- | --- | --- | --- |
| Blanchet S | 2024 | Dataset for Fargeot et al_Genetic diversity affects ecosystem functions across trophic levels as much as species diversity, but in an opposite direction | https://doi.org/10.6084/m9.figshare.25392496.v1 | figshare, 10.6084/m9.figshare.25392496.v1 |

The following previously published dataset was used:

| Author(s) | Year | Dataset title | Dataset URL | Database and Identifier |
| --- | --- | --- | --- | --- |
| Fargeot L | 2023 | Revealing genomic and species diversity patterns across multiple trophic levels in riverscapes | https://doi.org/10.6084/m9.figshare.21961358.v1 | figshare, 10.6084/m9.figshare.21961358.v1 |

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
