## [Editor Report · eLife Assessment]

This **important** study uses a comprehensive observational dataset to provide **solid** evidence on how genetic diversity and species diversity differentially affect multiple ecosystem functions within and across multi-trophic levels in an aquatic ecosystem. The work will be of interest to ecologists working on multi-trophic relationships and biodiversity.

---

## [Referee Report · Joint Public Reviews]

Summary:

This work used a comprehensive dataset to compare the effects of species diversity and genetic diversity on multiple ecosystem functions within each trophic level and across three trophic levels. The authors found that species diversity had negative effects on ecosystem functions, while genetic diversity had positive effects. These effects were only observed within each trophic level and not across the three trophic levels studied. Although the effects of biodiversity, especially genetic diversity across multi-trophic levels, have been shown to be important, there are still very few empirical studies on this topic due to the complex relationships and difficulties in obtaining data. This study collected an excellent dataset to address this question and improve our understanding of the effects of genetic diversity effects in aquatic ecosystems.

Strengths:

The study collected a large, good and rare observational dataset covering different facets of diversity (species vs. genetic, multi-trophic levels) and multiple ecosystem functions (biomass of focal species and overall communities, and decomposition rates). The authors used appropriate statistical analyses to provide a comprehensive analysis about how different facets of diversity affect different ecosystem functions.

Weaknesses:

The nature of this observational study makes it difficult to get compelling evidence of the causal relationships between biodiversity and ecosystem functions. As the ecosystem functions were measured at both species and community levels in natural ecosystems, particular care needs to be taken when interpreting comparisons between these ecosystem functions measured at different levels.

---

## [Author Response]

The following is the authors’ response to the previous reviews.

**eLife Assessment**
This important study provides empirical evidence of the effects of genetic diversity and species diversity on ecosystem functions across multi-trophic levels in an aquatic ecosystem. The support for these findings is solid, but a more nuanced interpretation of the results could make the conclusions more convincing. The work will be of interest to ecologists working on multi-trophic relationships and biodiversity.

Thanks for this new assessment. Here below we reply to the comments that you and the reviewer have made. We understand the critics related to the issue of the interpretation of causal relationships from observational data. We now added an entire paragraph (in the second paragraph of the Discussion) that explicitly call for a cautionary interpretation of our results. We also tried to refrain the use of certain words (e.g., “we demonstrate”) when we think it is hard to conclude. This a tricky exercise as on the one hand we gathered a large and strong database (which had been underlined by the reviewers) that should supposedly strengthen statistical inferences, but on the other hands, the inferences we’ve made are based from observational data, which obviously comes from biases (even if partially controlled statistically). We hope that you’ll find our adding appropriate to find the good balance between a strong dataset and fragile interpretation.

**Public Reviews:**

**Reviewer #1 (Public review):**
Summary:This work used a comprehensive dataset to compare the effects of species diversity and genetic diversity within each trophic level and across three trophic levels. The results stated that species diversity had negative effects on ecosystem functions, while genetic diversity had positive effects. Additionally, these effects were observed only within each trophic level and not across the three trophic levels studied. Although the effects of biodiversity, especially genetic diversity across multi-trophic levels, have been shown to be important, there are still very few empirical studies on this topic due to the complex relationships and difficulty in obtaining data. This study collected an excellent dataset to address this question, enhancing our understanding of genetic diversity effects in aquatic ecosystems.Strengths:The study collected an extensive dataset that includes species diversity of primary producers (riparian trees), primary consumers (macroinvertebrate shredders), and secondary consumers (fish). It also includes genetic diversity of the dominant species in each trophic level, biomass production, decomposition rates, and environmental data. The writing is logical and easy to follow.Weaknesses:The two main conclusions-(1) species diversity had negative effects on ecosystem functions, while genetic diversity had positive effects, and (2) these effects were observed only within each trophic level, not across the three levels-are overly generalized. Analysis of the raw data shows that species and genetic diversity have different effects depending on the ecosystem function. For example, neither affected invertebrate biomass, but species diversity positively influenced fish biomass, while genetic diversity had no effect. Furthermore, Table S2 reveals that only four effect sizes were significant (P < 0.05): one positive genetic effect, one negative genetic effect, and two negative species effects, with two effects within a trophic level and two across trophic levels. Additionally, using a P < 0.2 threshold to omit lines in the SEMs is uncommon and was not adequately justified. A more cautious interpretation of the results, with acknowledgment of the variability observed in the raw data, would strengthen the manuscript.

There is actually no objective justification for having chosen p<0.20. This is a subjective threshold that has been chosen to simplify the visual interpretation of causal graphs while highlighting the most biologically relevant links. We have now added a sentence stating explicitly the subjective nature of the threshold. We understand the point you raised regarding the cautionary interpretation of the results. We have now added a paragraph (just before the detailed discussion) explicitly calling for a cautionary interpretation of the results (see l. 414-424). We think this paragraph prevails for the entire discussion. Our message in this paragraph is that inferences that we’ve made can arise from both a biological reality and statistical artefacts. We can not really tease apart at this stage, and our interpretation of the results therefore has to be taken with care. We hope you’ll find the statement adequate. We prefer advertising the readers from the start rather than including cautionary note all over the discussion. We feel it was more logical and comfortable. We have also modified the text from place to place to avoid strong statement such as “we demonstrated” when we think the demonstration can not be considered as solid.

**Recommendations for the authors:**

**Reviewing Editor:**
In addition to the comments from the reviewer, we have the following comments on your paper:(1) It would be important to clarify that there could be different interpretations about one of the major findings: for within-trophic BEF relationships, genetic and species diversity have the opposite effects on ecosystem functions (i.e., positive and negative effects for genetic and species diversity, respectively). (1) One possibility is that for each specific ecosystem function, genetic and species diversity have the opposite effects. (2) The other possibility is that genetic diversity has positive effects on some functions, while species diversity has negative effects on other functions. These two possibilities can have quite different implications about the generalizability of the conclusion, mechanisms involved, and practices for ecosystem management. Therefore, it would be important to clarify that the findings from this paper are more about the second rather than the first possibility both in the discussion and conclusion sections.

Yes, true, this is an important distinction and we agree with your conclusion. We have added a section in the Discussion (l. 537-545) and a note in the Conclusion (l. 625-627).

(2) Please take special caution when comparing the findings from this observational study vs. previous experimental works. (1) The different ranges of diversity in the observational vs. experimental works, together with the nonlinear nature of the BEF relationship challenge the direct comparisons of their results. That is, even if their true BEF relationship are identical, focusing on different sections of a nonlinear curve can give us different results of the estimated BEF relationships. This challenge is further aggravated when involving both genetic and species diversity because these two facets have different biological meanings as the authors have already noted. Using standardized effect size or explained variance, as this paper did, may partially get around but not truly resolve this issue. It would be important to add clarifications to make the comparisons between genetic and species diversity effects more understandable in a biological or ecological context. One possibility could be to state that both genetic and species diversity measured in this study well represent their natural gradients in this aquatic ecosystem, so that the standardized effect sizes quantify how these natural diversity gradients associate with ecosystem functions. This further points to the issue about the representatives of the genetic diversity sampled from up to 32 individuals for each species per site, which would also need clarification. We suggest the authors to identify these challenges in the discussion, so that future studies can be aware of these or even find alternative solutions. (2) The species diversity effects have quite different meanings between this study and previous observational and experimental studies. The negative effects are for the biomass of one target species from this study, while the species diversity effects are usually for the biomass of all species within a community. These two scenarios are not directly comparable. The negative relationship between species diversity and a target species' biomass can simply arise from a sampling process, for example, given the same community biomass, the more species occur in a community, the less biomass allocated to a single species, without assuming any biological interactions or species differences. And this study cannot exclude this possibility. Note that this null, sampling process is not equal to a negative covariance between biomass of a focal species and biomass of the community involving the species as stated in lines 446-448. To avoid possible mis-interpretation, we suggest the authors to revise or remove the comparison appearing in the paragraph starting from line 515.

Thanks for these comments. Although we agree with the two points raised by the Editor, we must admit that we found them difficult to answer properly. See our detailed responses hereafter.

Point (1): this is true that comparisons with previous studies is tricky, especially when these comparisons also include both genetic and species components. This is a problem (a limit) for almost all comparisons in biology. We added a few lines to warn readers that these comparisons are not without any limits (see l. 414-424). Regarding the fact that « genetic and species diversity measured in this study well represent their natural gradients in this aquatic ecosystem »: all is about scales. The genetic and species diversity measured in this study are obviously representative of communities and populations of the upstream (piedmont) part of the Garonne River basin as our sampling design covers all the east-west gradient. On the other hand, these communities and populations are not representative of the entire Garonne River basin, as we lack all the downstream part of the network. We added a sentence to specify that the sampling communities are specific of this specific ecosystem (rivers from the piedmont, see l. 224-226). Regarding « the issue about the representatives of the genetic diversity sampled from up to 32 individuals », we must admit that we are surprised by this comment as it is a very classical way for estimating genomic diversity. Although there is no clear rule, 30 individuals per site is generally assumed (and has been shown) to be an appropriate sample size (especially given that we used here a genome-wide approach). We added a reference to justify the sample size.

Point (2): We understand the point raised by the Editors. Regarding your note “Note that this null, sampling process is not equal to a negative covariance between biomass of a focal species and biomass of the community involving the species as stated in lines 446-448.”: this is true, we rephrase this sentence to be more neutral. Regarding the paragraph starting l. 515 (now 550), we refrained to remove this paragraph as it provides some mechanistic explanation for underlying patterns, which we think is important even if incomplete or speculative. The confusion probably arises because here we discuss all type of negative BEFs, including the effect of species diversity on the biomass of the community, on the biomass of focal species (including those from other trophic levels) and the litter degradation. Our discussion is very general, whereas you seem to focus on a specific case of negative species-BEFs. To highlight this further and warn readers about possible conclusions, we added the following sentence: “Given the empirical nature of our study and the fact that our meta-regressive approach includes several types of BEFs (e.g., species richness acting either on the biomass of a single focal species or on the biomass of an entire focal community), it is hard to tease apart specific and underlying mechanisms” (l. 573-576).

(3) Please clarify how you derived the 95% CI in Fig. 5. For example, how did you involve the uncertainties of each raw effect size (e.g. each black triangle in Fig. 5a) when calculating their mean and 95% CI in each group (e.g., the red triangles and error bars in Fig. 5a)?

Estimates and 95%-CI from Figure 5 are derived from the mixed-effect models described from l. 314. They are hence marginal effects derived from the models, and 95%-CI include all error terms (fixed and random). We now specify in the Figure caption that estimates and 95%-CI are marginal effects derived from the mixed-effect models.